# An Immunological Polysaccharide from *Tremella fuciformis*: Essential Role of Acetylation in Immunomodulation

**DOI:** 10.3390/ijms231810392

**Published:** 2022-09-08

**Authors:** Tzu-Yin Huang, Feng-Ling Yang, Hsiao-Wen Chiu, Hong-Chu Chao, Yen-Ju Yang, Jyh-Horng Sheu, Kuo-Feng Hua, Shih-Hsiung Wu

**Affiliations:** 1Institute of Biological Chemistry, Academia Sinica, Taipei 115, Taiwan; 2Department of Biotechnology and Animal Science, National Ilan University, Ilan 260, Taiwan; 3Department of Marine Biotechnology and Resources, National Sun Yat-sen University, Kaohsiung 804, Taiwan

**Keywords:** *Tremella fuciformis*, polysaccharide, structure elucidation, acetylation, immunomodulatory activities, toll-like receptor 4

## Abstract

The edible fungus *Tremella fuciformis* was shown to have a high molecular weight (1.87 × 10^3^ kDa) bioactive polysaccharide, denoted as TFP-F1. Monosaccharide composition and NMR analysis of the polysaccharide and its derivatives indicated it contained fucose (Fuc*p*), xylose (Xyl*p*), mannose (Man*p*), and glucuronic acid (GlcA*p*) in a ratio of 0.9:1.0:3.2:1.2. Using IR, NMR, and GC-MS spectroscopic data, the structure of TFP-F1 was elucidated as {→3)-[*β*-D-GlcA*p*-(1→2)]-*α*-D-Man*p*-(1→3)-*α*-D-Man*p*-(1→3)-[*α*-L-Fuc*p*-(1→2)-*β*-D-Xyl*p*-(1→2)]-*α*-D-Man*p*-(1→}_n_, with partial acetylation of C6-OH in mannoses. Furthermore, at a concentration of 1 μg/mL, TFP-F1 was found to stimulate the secretion of TNF-α and IL-6 in J774A.1 macrophage cells in vitro via interaction with toll-like receptor 4 (TLR4). The removal of *O*-acetyl groups led to the loss of immunomodulatory activities, demonstrating that *O*-acetyl groups play an essential role in enhancing the production of pro-inflammatory cytokines.

## 1. Introduction

Mushrooms have been used in traditional cuisines and medicines for hundreds of years. Polysaccharides are regarded as one of the most critical ingredients in these fungi and have been shown to exhibit different biological functions [1]. For example, mushroom *β*-1,3-glucans have immunomodulatory activities [2,3,4]. In addition, a *β*-1,3-glucan with 1,6-branches, lentinan, produced from the edible fungus *Lentinus edodes*, was also reported to display potential antitumor activities [5].

The mushroom *Tremella fuciformis*, also called snow ear or white jelly mushroom, belonging to the Tremellaceae family of Basidiomycota fungi, is a traditional delicacy in Asia. The chemical composition of *T. fuciformis* contains carbohydrates (70–80%, *w*/*w*), proteins (8–10%, *w*/*w*), fibers (2–3%, *w*/*w*), and trace amounts of lipids [6,7]. It is considered to be an “elixir” and thus has high economic value [8]. In previous studies, polysaccharides from *T. fuciformis* demonstrated a range of biological activities, including acting as anti-aging compounds [9], antitumor reagents [10], restorations of memory impairment [11], hypoglycemic agents [12], causes of hypocholesterolemia [13], and as immunomodulatory molecules [14]. Polysaccharides from *T. fuciformis* have been reported to be mainly composed of D-mannose, D-xylose, and D-glucuronic acid [8]. However, their high molecular weight and complexity have led to difficulties in collecting NMR spectroscopic data to establish their complete structure. Detailed structural elucidation of these polysaccharides using NMR spectroscopic analysis has not been reported. Instead, structural studies of these polysaccharides are only based on monosaccharide composition and linkage analysis by GC-MS [15,16,17,18,19,20].

Macrophages of the innate immune system are the first line of defense against various microorganisms [21]. Toll-like receptors (TLR) play an essential role as receptors for microbial components and for detecting invading pathogens or other foreign materials [22]. It is known that LPS could stimulate macrophages via TLR4 to produce different pro-inflammatory cytokines. Also, TLR4 has been demonstrated to be a receptor of some polysaccharides [23,24], including the polysaccharides isolated from *Auricularia auricula-judae* and *Antrodia cinnamomea* stimulate the production of pro-inflammatory cytokines by triggering the TLR4/MD2 pathway [25,26,27]. Cytokines produced by macrophages can be used to evaluate the immunomodulatory activity of polysaccharides [28].

More detailed structure–activity relationships (SAR) are critical to understanding the biological activity of polysaccharides and prospect applications [29]. Obtaining an accurate polysaccharide structure is a necessary prerequisite to carrying out SAR studies. Also, the detailed SAR investigation of *Tremella fuciformis* polysaccharide has not been studied. Therefore, the aim of this study was to establish the whole structure of a bioactive *T. fuciformis* polysaccharide; other goals were to understand its immunomodulatory activity and working mechanism. We describe here the isolation and structural determination of a bioactive polysaccharide (TFP-F1) from *T. fuciformis*. The structural work was facilitated by the degradation of the full-length polysaccharide with the FeSO_4_/ascorbic acid (Vc)/H_2_O_2_ method [30]. In addition, the immunomodulatory activities of TFP-F1 and its derivatives and critical structural motifs necessary for this immunomodulation were also investigated.

## 2. Results and Discussion

### 2.1. Extraction, Purification, and Determination of the Molecular Weight of TFP-F1

As shown in Figure 1a, hot water extraction of dried fruiting bodies of *T. fuciformis* (200.4 g), followed by precipitation with absolute ethanol, led to the isolation of a white crude polysaccharide (278.3 mg). This crude product was purified by treating with deoxyribonuclease I (DNase I), ribonuclease (RNase), and proteinase K to remove nucleic acids and proteins. The FeSO_4_/Vc/H_2_O_2_ degradation method was applied to the reactants to increase the water solubility of the polysaccharide and decrease the viscosity of solutions containing it, thus providing, after dialysis, polysaccharide TFP (166.1 mg). Finally, TFP was further purified through an HW-65F size exclusion column to obtain the pure polysaccharide TFP-F1 (118.6 mg) (Figure 1b). The molecular weight of TFP-F1 was calculated based on the standard curve (Figure 1c) and determined to be 1.87 × 10^3^ kDa (Figure 1d).

### 2.2. Monosaccharide Composition and Linkage Analysis of TFP-F1 and Its Derivatives

The *Tremella* polysaccharides were reported to contain D-glucuronic acid residues [31]. In our experimental procedure of monosaccharide composition, the resulting products containing hexauronic acid could not be detected by GC-MS directly if it was not reduced. As such, the derivation of TFP-F1 was carried out. TFP-F1 and its derivatives were subjected to methanolysis, acetylation, and trimethylsilyation to afford derivatives for GC-MS analysis. This analysis indicated that TFP-F1 was composed of L-fucose, D-xylose, and D-mannose in a ratio of 1.0:1.2:2.9 (Appendix A). Reduction of the uronic acid residues in TFP-F1 was carried out by treatment with 1-cyclohexyl-3-(2-morpholino-ethyl) carbodiimidemetho-*p*-toluenesulfonate (CMC) and NaBH_4_, resulting in a reduced polysaccharide TFP-F1R, which was shown to contain L-fucose, D-xylose, D-mannose, and D-glucose in a ratio of 0.9:1.0:3.2:1.2 (Appendix A). An increase in the proportion of glucose compared with the analysis performed before the carboxyl group reduction indicated the presence of terminal glucuronic acid residue in the native polysaccharide TFP-F1. Further, deacetylation of TFP-F1R was achieved by treatment with 0.1 N NaOH for 24 h, giving a product, deOAc TFP-F1R. The ^1^H NMR spectrum of deOAc TFP-F1R (Appendix A) revealed the disappearance of the signal at *δ* 2.10 ppm, compared to the NMR spectrum of TFP-F1, indicating the deacetylation was successful. In addition, the derivative, TFP-F1P, was prepared using the periodate cleavage of TFP-F1. This sample only consists of D-mannose (Appendix A).

A modified Hakomori method, Ciucanu/Kerek methylation [32,33], was applied to prepare partially methylated acetylated alditols for linkage analysis. These results indicated that TFP-F1 possessed 2,3-linked-D-mannopyranose residues, 3-linked-D-mannopyranose residues, 2-linked-D-xylopyranose residues, and terminal-6-deoxy-L-galactopyranose (Fuc) residues (Appendix A). Furthermore, one additional terminal-D-glucopyranosyl residue appeared in TFP-F1R, indicating that the glucuronic acid residue was located in the terminal position of the structure (Appendix A). Based on these data, the structure of TFP-F1 was deduced to contain 1,2,3-linked-mannose, 1,3-linked-mannose, 1,2-linked-xylose, and terminal fucose. In addition, TFP-F1P was demonstrated to have only 1,3-linked mannose (Appendix A), indicating the backbone of TFP-F1 is a linear 1,3-linked mannan.

### 2.3. Structural Elucidation of TFP-F1 and Its Derivatives by IR and NMR Spectra

The IR spectrum (Appendix A) of TFP-F1 showed absorbance at 3366 and 2935 cm^−1^, corresponding to O–H and C–H stretching frequencies, respectively. Weak absorptions at 1647 and 1418 cm^−1^ suggested the presence of asymmetric and symmetric COO^–^ stretches, respectively [34]. The absorption at 1370 cm^−1^ arises from the characteristic bending of methyl groups. Further, the strong absorption band range from 1070 to 1035 cm^−1^ was dominated by the ring and side groups of vibrational mode [35].

NMR spectra of TFP-F1 (Appendix A) were recorded on Bruker AVIII-800 NMR spectrometer at 323 K. The ^1^H NMR spectrum (Appendix A) showed six signals (*δ* 4.37, 4.44, 5.06, 5.10, 5.13, and 5.47 ppm) in the anomeric region, which were designed as residues **A**, **B**, **C**, **D**, **E**, and **F**, respectively. Based upon correlations in the HSQC spectrum (Appendix A), the ^13^C chemical shifts in the anomeric region (Appendix A) for residues **A**–**F** were assigned to be *δ* 101.7, 102.2, 101.3, 102.3, 100.7, and 97.4 ppm, respectively.

The ^1^H anomeric chemical shift of residue **A** was assigned as *δ* 4.37 ppm. Analysis of the COSY spectrum (Appendix A) showed the H-2 of residue **A** was at *δ* 3.50 ppm. Further analysis of COSY and TOCSY spectra (Appendix A) suggested the assignments of H-3 and H-4 to be *δ* 3.59 and *δ* 3.55 ppm, respectively. Additionally, HMBC correlations of **A**H-1 to **A**C-5 and **A**H-2 to **F**C-6 were observed. Based on this evidence and comparing the ^1^H and ^13^C chemical shifts with previous reports [36], residue **A** was identified as a 1,2-linked *β*-xylopyranose. With the HSQC, COSY, and TOCSY spectra, the correlations of H-1 to H-5 were assigned as *δ* 4.44, 3.33, 3.41, 3.53, and 3.65 ppm, respectively, in residue **B**. Furthermore, HMBC correlations (Appendix A) of both **B**H-4 and **B**H-5 to **B**C-6 (*δ* 175.7 ppm) were present, suggesting that residue **B** was a terminal-D glucuronic acid residue.

The ^1^H anomeric signal of residue **C** was assigned as *δ* 5.06 ppm, which showed a correlation to C-2 (*δ* 78.1 ppm) in the HMBC spectrum (Appendix A). From the HSQC spectrum (Appendix A), the ^1^H chemical shift of H-2 was assigned as *δ* 4.15 ppm. Furthermore, an HMBC correlation between **C**H-1 to **C**C-3 (*δ* 76.1 ppm) was observed. Downfield ^13^C resonances of both C-2 and C-3 compared with the standard values [36] suggested residue **C** is a 1,2,3-linked *α*-mannose. Likewise, detailed analysis of COSY, TOCSY, and HMBC correlations, residues **D** and **E** were determined to be a 1,3-linked *α*-mannose and a 1,2,3-linked *α*-mannose, respectively.

The monosaccharide composition analysis of TFP-F1 (Appendix A) indicated it was composed of fucose, xylose, and mannose in a ratio of 1.0:1.2:2.9. Therefore, residue **F**, which possessed a downfield ^1^H anomeric signal at *δ* 5.47 ppm, was confirmed to be a terminal L-fucose. The ring proton signals of residue **F** were assigned via the COSY, TOCSY, and HMBC correlations. In the COSY and TOCSY spectra (Appendix A), correlations of H-1 to H-3 were observed. Therefore, the ^1^H resonances of H-2 and H-3 were assigned as *δ* 3.72 ppm and 3.78 ppm, respectively. The remaining signals were determined by HMBC correlations. The HMBC correlations of **F**H-6 (*δ* 1.15 ppm) to **F**C-5 (*δ* 66.6 ppm) and **F**C-4 (*δ* 72.2 ppm) were found (Appendix A). Based on the above evidence, the ^1^H and ^13^C chemical shifts of residue **F** were confirmed (Table 1 and Appendix A). Other NMR spectra, including COSY and selective HMBC correlation of TFP-F1 (Appendix A), ^1^H NMR and ^13^C NMR of reduced TFP-F1 (named TFP-F1R) and de-*O*-acetylated of reduced TFP-F1R (named deOAc TFP-F1R) were also analyzed respectively (Appendix A).

The linkage of each residue was elucidated by the assignment of correlations in the HMBC spectrum (Appendix A). HMBC correlations of **C**H-1 to **E**C-3, **E**H-1 to **D**C-3, and **D**H-1 to **C**C-3 were found, suggesting the backbone chain of TFP-F1 is an *α*-1,3-linked mannan. Furthermore, HMBC correlations of **A**H-1 to **C**C-2 and **B**H-1 to **E**C-2 indicated both the *β*-D-xylopyranose and *β*-D-glucopyranosiduronic acid residues were attached to the C-2 positions of residue **C** and residue **E**, respectively. In addition, the presence of HMBC correlations of **F**H-1 to **A**C-2 and **A**H-2 to **F**C-1 indicated that the *α*-L-fucopyranose was attached to the C-2 position of *β*-D-xylopyranose. Accordingly, the chemical structure of TFP-F1 was elucidated as {→3)-[*β*-D-GlcA*p*-(1→2)]-*α-*D-Man*p*-(1→3)-*α-*D-Man*p*-(1→3)-[*α-*L-Fuc*p*-(1→2)-*β-*D-Xyl*p*-(1→2)]-*α-*D-Man*p*-(1→}_n_. In addition, the COSY correlation (Appendix A) of *δ* 4.31 and *δ* 4.34 ppm was found. From the HSQC spectrum (Appendix A), the ^13^C chemical shift corresponding to both signals was assigned as *δ* 63.7 ppm. Further, the HMBC correlations (Appendix A) of both signals to an ester group (*δ* 174.3 ppm) were observed, suggesting that the position with partial acetylation should be C-6. Furthermore, the HMBC correlations (Appendix A) of **C**H-5 (*δ* 3.75 ppm), **D**H-5 (*δ* 3.78 ppm), and EH-5 (*δ* 3.78 ppm) to *δ* 63.7 ppm were found. By comparing the chemical shifts with previous reports [37], mannose residues were deduced to be partially acetylated at the C-6 position (Figure 2).

### 2.4. TFP-F1 Enhances the Production of Pro-Inflammatory Cytokines through Interaction with TLR4

The immunomodulatory activities of TFP-F1 were investigated by detecting the secretion of pro-inflammatory cytokines in J774A.1 macrophage cells. Polymyxin B was used to suppress any LPS contamination. As shown in Figure 3a, polymyxin B treatment strongly inhibited LPS-induced TNF-*α* production. At the same time, only a trace amount of TNF-*α* reduction was observed upon treatment with TFP-F1 and polymyxin B. Furthermore, TFP-F1 and its derivatives were quantified whether LPS existed was detected by using Limulus Amebocyte Lysate (LAL) Assay. The results (Appendix A and Appendix A) indicated that each sample (50 μL) at a concentration of 1 μg/mL contained less than 0.2 endotoxin units (EU), indicating that TFP-F1 was relative endotoxin-free (<0.1%). Meanwhile, the mild reduction in TNF-α is possibly caused by the charge effect between polymyxin B and carboxylic acid of GlcA*p* or *O*-acetylation of mannoses. These results showed that TFP-F1 could enhance the production of TNF-*α* in J774A.1 macrophage cells, even at a low concentration of 1 μg/mL (Figure 3b). In addition, to confirm that the sample possessed cytotoxicity against the normal cell, J774A.1 macrophages were treated with different concentrations of TFP-F1 (including 40, 20, 10, and 5 μg/mL) for three days. The viability of the macrophage cells was measured using Alamar blue assay (Appendix A). These results (Appendix A) showed the viability of macrophage cells is more than 85%, indicating the TFP-F1 sample is non-toxic.

TLR4 is a well-known receptor of polysaccharides. Lipopolysaccharide, located on the outer membrane of Gram-negative bacteria, is one of the ligands of TLR4. Its toxic component, lipid A, was reported to bind with the MD2 hydrophobic pocket and connect the dimer of TLR4s [38]. The activation of TLR4 could recruit the cytoplasmic adaptor protein TIRAP (Toll/IL-1R domain-containing adaptor protein) and MyD88 (myeloid differentiation primary response gene 88) proteins to form a TLR complex, which led to the induction of pro-inflammatory cytokines, such as TNF-α and IL-6 [39,40]. However, fungus polysaccharides without lipid A were reported to trigger the immune response [25,26]. Additionally, various polysaccharides with different structures were found to be a ligand of TLR4, suggesting that the configurations and components of polysaccharides could be considered to play essential roles in affecting TLR4 reaction [23].

To identify the putative receptor for TFP-F1, wild-type RAW-Dual^TM^ cells and RAW-Dual^TM^ KO-TLR4 cells were treated with LPS or different concentrations of TFP-F1, and the production of TNF-*α* and IL-6 cytokines were detected. Figure 3b,c shows a significant difference between the wild-type RAW-Dual^TM^ cells and the TLR4-knockout RAW-Dual^TM^ cells. Human TLR4-expressing HEK 293 cells were also incubated with LPS or TFP-F1, and the production of TNF-*α* and IL-6 was followed. As shown in Figure 3d,e, compared to the wild-type human HEK 293 cells, the TLR4-expressing cells displayed an apparent increase in cytokine production. Based on the above evidence, we proposed that TFP-F1 is a TLR4 ligand.

### 2.5. Immunomodulatory Activities of TFP-F1 Derivatives

TFP-F1 had been demonstrated to display significant immunomodulatory activity and enhance the secretion of TNF-*α* in the J774A.1 macrophage cell. Structure elucidation of TFP-F1 indicated it possessed carboxylic acid and *O*-acetyl groups. To better understand the structural motifs of TFP-F1 that induce the immune response, derivatives of TFP-F1 were prepared for evaluation (Figure 4a). A carboxyl-reduced sample (named TFP-F1R) was prepared by reducing TFP-F1 with NaBH_4_, while deOAc TFP-F1R was afforded by treating TFP-F1R with sodium hydroxide. To determine the content of the *O*-acetyl group, the ^1^H NMR spectra of TFP-F1 and deOAc TFP-F1R were recorded (Appendix A). TFP-F1P is the product resulting from the periodate cleavage of TFP-F1. It was found that TFP-F1P (*α*-1,3-mannan) did not lead to the production of cytokines in the J774A.1 macrophage cells, suggesting that the branched structure on the 1,3-mannan of TFP-F1 might be essential for immunomodulatory activity (Figure 4b). Moreover, as shown in Figure 4c, TFP-F1R could enhance the secretion of cytokines in J774A.1 macrophage cells at a level comparable with TFP-F1, indicating that the GlcA*p* motif, or at least its carboxylic acid moiety, was not essential for activity. On the other hand, strikingly, deacetylation of TFP-F1 led to essentially complete loss of TNF-*α* production. As such, the *O*-acetyl group in TFP-F1 is essential for immunomodulatory activity. The relationship between the structures and immunomodulatory activities of TFP-F1 derivatives was in agreement with our previous research [26,27,41].

Previous literature indicated that the activation of NF-κB required acetylation [42]. Also, acetylation has been proven to play an essential role in the TLR4/NF-κB signaling pathway for the secretion of pro-inflammatory cytokines [43]. As such, we used human TLR4-expressing HEK 293 cells treated by the TFP-F1 sample to measure the activation level of NF-κB. As shown in Figure 5, the TLR4-expressing cells showed a noticeable increase in NF-κB activation. Based on the result, we demonstrated that the TFP-F1 could stimulate the production of pro-inflammatory cytokines through the TLR4/NF-κB signaling pathway. We suggested the crucial reasons for the *O*-acetyl group, which might be related to the non-polar interaction with the π system in the TLR4-MD2 complex [24]. At present, various exogenous and endogenous ligands which triggered TLR4 have been found. These ligands possessed a certain degree of hydrophobicity that might be buried inside the MD2 pocket, contributing to interaction with TLR4 [44]. Accordingly, the *O*-acetyl group in TFP-F1 might affect the immunomodulatory activities.

## 3. Materials and Methods

### 3.1. Preparation of Polysaccharides (TFP-F1) from T. fuciformis

A dried sample of *T. fuciformis* (200.4 g) was extracted with 95% aqueous ethanol (*v*/*v*) for 24 h to remove small polar molecules. The residue (36.5 g) was then extracted with hot water (1 L) for 1 h at 100 °C. The supernatant was precipitated by the addition of 80% aqueous ethanol (*v*/*v*) to afford a crude polysaccharide (278.3 mg). For further purification, the crude polysaccharides were treated with deoxyribonuclease I (Sigma-Aldrich, St. Louis, MI, USA) and ribonuclease (Sigma-Aldrich, St. Louis, MI, USA) at 37 °C for 4 h, followed by treatment with proteinase K (Bioshop, Burlington, ON, Canada) under the same conditions for 24 h to remove nucleic acids and proteins. The resulting product was dialyzed (10 kDa cell membrane) against deionized water to afford a purified polysaccharide, TFP (166.1 mg). Due to the low water-solubility of TFP and high viscosity of its solutions, degradation using the FeSO_4_/Vc/H_2_O_2_ method was carried out [30]. Thus, TFP was treated with a mixture containing 2 mM FeSO_4_, 5 mM H_2_O_2_, and 20 mM ascorbic acid at room temperature for 2 h. The reactants were dialyzed (10 kDa cell membrane) against deionized water for 48 h and then lyophilized to obtain the degraded sample. For further purification, size exclusion chromatography on an HW-65F column (Tosoh Bioscience, 10–1000 kDa, 1.6 cm D × 90 cm H) was carried out. The presence of carbohydrates in column fractions was determined using the phenol–sulfuric acid method [45] and afforded pure TFP-F1 (118.6 mg).

### 3.2. Molecular Weight Determination of TFP-F1

The molecular weight (Mw) of TFP-F1 was determined by high-performance size exclusion chromatography (Agilent 1100 HPLC VWD System, Agilent Co., Palo Alto, CA, USA). The system contained a G1310A isocratic pump, a G1322A degasser, a G1314A variable wave-length detector, a G1362A refractive index detector, and a BioBasicTM SEC-1000 size exclusion chromatography HPLC column (7.8 mm D × 300 mm L; 5 μM particle size). The sample of TFP-F1 (1.0 mg) was dissolved in 1 mL of deionized water and then chromatographed at a flow rate of 1 mL/min.

### 3.3. Periodate Cleavage of TFP-F1

TFP-F1 was treated with sodium periodate to further fragment the polymer following established methods [46]. Briefly, TFP-F1 (21.9 mg) was dissolved in deionized water (12 mL), and 30 mM NaIO_4_ (12 mL) was further added. The solution was kept in the dark for three days until the optical density value at 223 nm became stable. Subsequently, the reactants were dialyzed using a 1 kDa membrane against deionized water, and then the product was reduced with 2M NaBH_4_ solution for 24 h. Finally, the resulting products were neutralized by 50% acetic acid (1 mL) to afford the final product, TFP-F1P (15.7 mg, 71.7%).

### 3.4. Carboxyl Reduction of the Hexauronic Acid in the TFP-F1

TFP-F1 (7.0 mg) was dissolved in 5 mL of deionized water, followed by the addition of 100 mg of CMC (Sigma-Aldrich, USA). The reaction was stirred for 2 h in an ice bath. During this period, the pH value was maintained at 4.7 by adding 0.1 N hydrochloric acid. To decrease the decomposition of the NaBH_4_ in the following step, 1 mL imidazole (HCl) buffer (4 M, pH = 7.5) was added [47]. Next, two batches of 100 mg NaBH_4_ were added slowly, and the solution was stirred in an ice bath for 2 h. The final products were dialyzed (10 kDa cell membrane) against deionized water over 48 h, lyophilized, and purified by size exclusion chromatography on Bio-Gel P-6 to afford the reduced sample TFP-F1R (3.5 mg, 50%).

### 3.5. Deacetylation of TFP-F1R

TFP-F1R (1.5 mg) was deacetylated by incubation with 0.1 N NaOH at 25 °C for 24 h. The resulting product was dialyzed (10 kDa membrane) against deionized water for 48 h and then lyophilized to obtain the deacetylated products, denoted as deOAc TFP-F1R (1.2 mg, 80%).

### 3.6. Monosaccharide Composition and Linkage Analysis

TFP-F1 and its derivatives (approximately 1 μg) in a glass tube were treated with 200 μL 0.5 M methanolic HCl (Sigma-Aldrich, USA) at 84 °C for 16 h. After cooling to room temperature, the reactants were evaporated under a flow of N_2_ gas, and then the acetylation by treatment with a solution containing 500 μL methanol, 50 μL acetic acid, and 10 μL pyridine for 20 min. After drying by the passage of N_2_ gas over the sample, the reactants were treated with 200 μL of a trimethylsilylation reagent (HMDS/TMCS/pyridine, 2:1:10, Sigma-Aldrich, USA) for 30 min. After drying by N_2_, the resulting products were then dissolved in 400 μL *n*-hexane (GC-MS grade) for analysis on a Bruker SCION SQ gas chromatography-mass spectrometry (GC-MS) using a DB-5MS fused silica capillary column (30 m, 0.25 mm i.d., 0.25 μm film). The monosaccharide composition of the polysaccharide was analyzed by comparing the retention time of the peaks with those of monosaccharide standards, including L-rhamnose, D-xylose, L-fucose, D-mannose, D-galactose, and D-glucose. The peak area of each sugar was calculated through the GC-MS data analysis software.

For the glycosidic linkage analysis of TFP-F1 and its derivatives, a modified Hakomori method, Ciucanu/Kerek methylation [32,33], was conducted. Approximately 2.0 mg sample was added to a glass tube and then dissolved in 0.5 mL dimethyl sulfoxide. Subsequently, 1–2 pestles of pulverized sodium hydroxide were added, and the solution was kept for 3 h before treatment with 0.5 mL methyl iodide. After 3 h, the reactants were partitioned at least four times between 2 mL chloroform and 2 mL deionized water, and then the combined organic layers were evaporated under a flow of N_2_ gas. The methylated product was hydrolyzed using 2 M 200 μL trifluoroacetic acid at 100 °C for 6 h and then, after cooling to room temperature, reduced using a 200 μL NaBD_4_ solution for 2 h. The excess NaBD_4_ was removed by glacial acetic acid. The reduced products were acetylated by adding 200 μL acetic anhydride and 100 μL pyridine at 80 °C for 1 h. The partially methylated acetylated alditols (PMAA) of TFP-F1 and its derivatives were dissolved in 400 μL *n*-hexane (GC-MS grade) for analysis by GC-MS.

### 3.7. IR and NMR Spectroscopic Data

The IR spectrum of TFP-F1 was recorded on FT-IR Tensor 27 (Bruker). This spectrum was recorded, ranging from 4000 cm^−1^ to 600 cm^−1^. The number of scans and resolution were set up with 64 scans at a resolution of 4 cm^−1^. The NMR spectra of TFP-F1 were recorded on the Bruker AVIII-800 instrument at 323 K. The TFP-F1 sample (7.1 mg) was dissolved in 500 μL deuterium oxide (99.5%) for data collection. The ^1^H chemical shifts of TFP-F1 were referenced to D_2_O at *δ* 4.49 ppm (323 K) as an internal standard. For further structure elucidation of TFP-F1, ^1^H–^13^C, correlation (COSY), total correlation (TOCSY), heteronuclear single quantum coherence (HSQC), heteronuclear multiple bond correlation (HMBC), and nuclear Overhauser effect (NOESY) spectra were obtained. Furthermore, the ^1^H NMR and ^13^C spectra of TFP-F1R and deOAc TFP-F1R were recorded on the same instrument at room temperature. The ^1^H chemical shifts of both above samples were referenced to D_2_O at *δ* 4.70 ppm (298 K) as an internal standard.

### 3.8. Cell Line Culture

The J774A.1 murine macrophage cell line was obtained from American Type Culture Collection (Rockville, MD, USA). Wild-type RAW-Dual^TM^, RAW-Dual^TM^ KO-TLR4, HEK-Blue hTLR4, and HEK-Blue Null2 cell lines were purchased from InvivoGen (San Diego, CA, USA). All cells were cultured in RPMI-1640 medium with 10% FBS and then maintained at 37 °C under a humidified atmosphere of 5% CO_2_ in an incubator.

### 3.9. Cell Viability

The J774A.1 murine macrophage cells (1 × 10^4^ cells/well) were cultured in a 96-well plate overnight. Subsequently, the cells were treated with different concentrations (5 μg/mL, 10 μg/mL, 20 μg/mL, and 40 μg/mL) of TFP-F1 and incubated at 37 °C under a humidified atmosphere of 5% CO_2_ in an incubator for 24 h. Then alamarBlue™ Cell Viability Reagent (10 μL) was added to each well and incubated for 2 h under the same condition. Cell viability was measured at the absorbance of 570 nm.

### 3.10. Cytokine Measurement

TFP-F1 and its derivatives (1.0 mg) were dissolved in 1 mL of deionized water to prepare a stock solution. J774A.1 macrophage cells were cultured in RPMI-1640 medium supplemented with 10% fetal bovine serum (FBS) and maintained in the condition of 5% CO_2_ at 37 °C. Briefly, J774A.1 macrophage cells (1 × 10^5^ cells/well) were cultured in a 24-well plate overnight, followed by the treatment with lipopolysaccharide (LPS) (1 μg/mL) or different concentrations of TFP-F1 (5 μg/mL, 10 μg/mL, and 20 μg/mL) at 37 °C for 24 h to keep the optimum proliferation status for experiments. Wild-type RAW-Dual^TM^ cells and RAW-Dual^TM^ KO-TLR4 cells (2 × 10^5^ cells/well) were cultured under the same conditions, and then treated with LPS (1 μg/mL) or different concentrations (1 μg/mL, 3 μg/mL, and 5 μg/mL) of TFP-F1 for 24 h. The production of pro-inflammatory cytokines, including TNF-*α* and IL-6, was measured at the absorbance of 570 nm by ELISA according to the protocol provided by the manufacturer (Introgen Therapeutics, Inc., Austin, TX, USA).

### 3.11. Limulus Amebocyte Lysate (LAL) Assay

The LAL assay was performed to quantify the endotoxin contamination in the TFP-F1 and its derivatives. *E. coli* endotoxin as standard control (25 EU), lyophilized LAL, chromogenic substrate, and LAL reagent water were purchased from Lonza (Walkersville, MD, USA). Each sample (50 μL) with a concentration of 1 μg/mL was detected following the protocol provided by the manufacturer*’*s instruction.

### 3.12. Quanti-Blue Assay

HEK-Blue hTLR4 and HEK-Blue Null2 Cells (2 × 10^5^/well) were seeded in 24-well plates in 100 μL of complete medium and treated with LPS (1 μg/mL) or different concentration samples (1, 3, and 5 μg/mL) of TFP-F1. After 24 h of stimulation, NF-κB activation led to secreted embryonic alkaline phosphatase (SEAP) production in the medium, quantified by the QUANTI-Blue^TM^ colorimetric reagent according to the manufacturer’s protocol (InvivoGen, San Diego, CA, USA). HEK-Blue Null2 cells served as a parental control cell line of HEK-Blue hTLR4.

### 3.13. Statistical Analysis

The cell viability and immunomodulatory results are expressed as the mean ± standard error of the mean (SEM) of triplicate measurements. Each experiment was repeated at least three times to confirm the reproducibility of the findings. Differences between treatment means were considered statistically significant at *p* < 0.05.

## 4. Conclusions

As an edible fungus with high economic value, *T. fuciformis* is rich in proteins, polysaccharides, and dietary fiber. Polysaccharides from this organism have been studied for nearly 50 years [36]. However, the physical characteristics (high molecular weight and high viscosity of polysaccharide solutions) have limited research progress [30]. In the present study (Figure 6), we used a reported extraction method [48] and further purification (including precipitation, dialysis using 8–10 kDa membrane, and size-exclusion chromatography) to afford TFP-F1, a bioactive polysaccharide from *T. fuciformis*, in which proteins, nucleic acids, smaller molecules, and peptides were removed. The sugar composition and linkage analysis of this polysaccharide and its derivatives were investigated by GC-MS. The results indicated that this polysaccharide, TFP-F1, was composed of L-fucose, D-xylose, D-mannose, and D-glucuronic acid in a ratio of 0.9:1.0:3.2:1.2. Detailed analysis of the NMR and GC-MS spectroscopic data provided the whole structure of TFP-F1 as {→3)-[*β*-D-GlcA*p*-(1→2)]-*α-*D-Man*p*-(1→3)-*α-*D-Man*p*-(1→3)-[*α-*L-Fuc*p*-(1→2)-*β-*D-Xyl*p*-(1→2)]-*α-*D-Man*p*-(1→}_n_, C6-OH of mannose was partially acetylated.

Mannose-containing polysaccharides are one of the main components of the fungal cell wall. At present, different mannan polysaccharides have been demonstrated to show nutritional and therapeutic values [49]. Generally, monosaccharides of mannose with *α*-(1,3) or *β*-(1,4) linkage are frequently displayed in the TLR4-activated polysaccharides. As such, they would be considered an essential backbone of TLR4 binding [23,24]. The immunomodulatory activity of TFP-F1 and its derivatives were also assayed. It was found that, at a concentration of 1 μg/mL, TFP-F1 could significantly enhance the production of TNF-*α* and IL-6 in vitro through interaction with TLR4. By comparing the immunomodulatory activity of various TFP-F1 derivatives, we confirmed that the *O*-acetyl group plays a vital role in immunomodulation. This result is consistent with our previous study on the polysaccharides from *Auricularia auricula-judae* [26,27], which revealed the importance of an *O*-acetyl group to immunomodulatory activity. Based on the above description, TFP-F1 could be considered a potential immunostimulant.

Until now, polysaccharides isolated from *Tremella fuciformis* have been demonstrated to have anti-oxidative, anti-inflammatory, and anti-aging effects [50]. Additionally, it has been clinically used for antineoplastic agents as a safe and non-toxic natural active product [51]. Furthermore, in this study, the viability of the macrophage cells treated with TFP-F1 was detected using Alamar blue assay, and the results also indicated that this sample is non-toxic. We plan to study further the potential of endotoxin tolerance by TFP-F1 to demonstrate its immunoprotective property against bacterial infection.

Strikingly, this is the first research to elucidate the chemical structure of *T. fuciformis* polysaccharides by the analysis of NMR spectroscopic data and to gain insight into the structural motifs in the polysaccharides important for immunomodulatory activity. The study of polysaccharide structures, immunomodulatory activities, and interaction with TLR4 will help the development of immunostimulants or adjuvant therapy [52]. Although many polysaccharides have been identified to be a ligand of TLR4, the investigation of oligosaccharides stimulated pro-inflammatory cytokines through TLR4 is rare. In the future, we will focus on the oligosaccharide repeating unit from TFP-F1 using synthetic methods and further evaluate whether this sample will enhance the immunomodulation through TLR4.

## Figures and Tables

**Figure 1 ijms-23-10392-f001:**
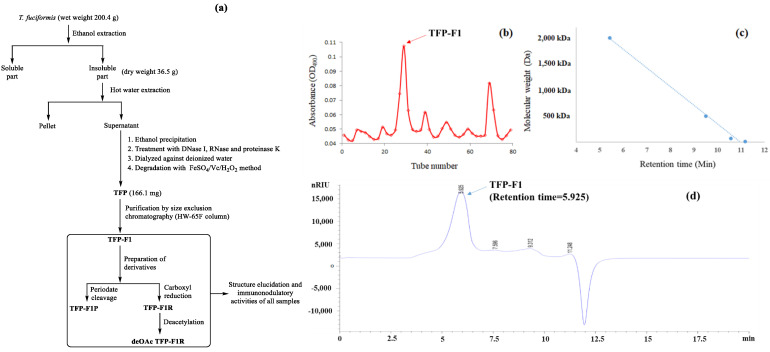
(**a**) A flow chart showing the procedures used to purify polysaccharides from *T. fuciformis* and evaluate their immunomodulatory activity (**b**) Size-exclusion chromatography HW-65F profile of TFP-F1. Fractions containing carbohydrates were determined using the phenol–sulfuric acid method. TFP-F1 was collected from tubes 26 to 33. (**c**) Standard curves of the dextran samples in HPLC, including 2000 kDa, 500 kDa, 70 kDa, and 10 kDa, were run in a BioBasic SEC-1000 column. (**d**) MW analysis of TFP-F1 was followed by (**c**).

**Figure 2 ijms-23-10392-f002:**
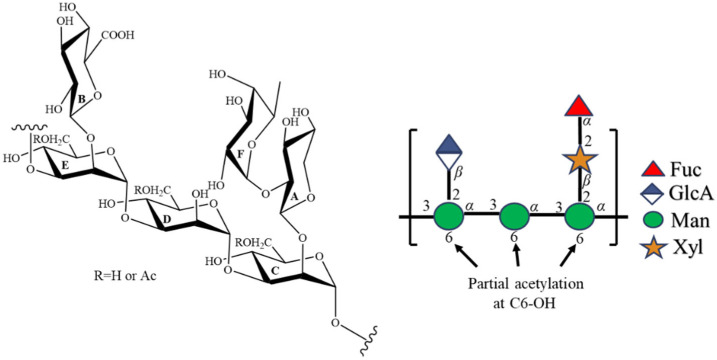
The chemical structure and symbolic nomenclature of TFP-F1.

**Figure 3 ijms-23-10392-f003:**
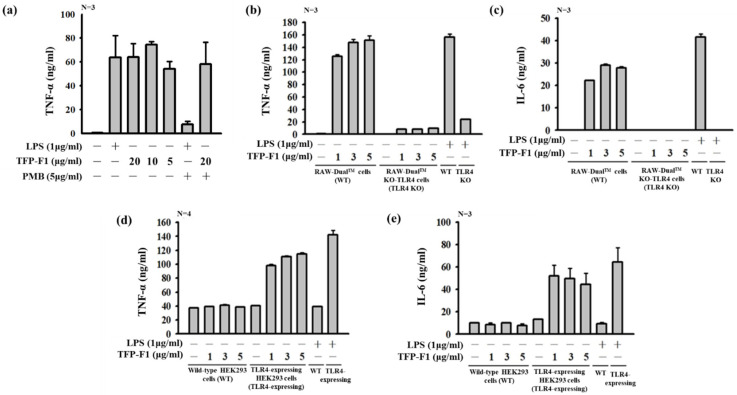
Immunomodulatory activity of TFP-F1. J774A.1 macrophage cells were incubated for 24 h with or without different concentrations of TFP-F1 or LPS (1 μg/mL). (**a**) Induction of TNF-*α* secreted by TFP-F1 (5, 10, and 20 μg/mL) in J774A.1 macrophage cells. (**b**,**c**) Wild-type RAW-Dual^TM^ cells and RAW-Dual^TM^ KO-TLR4 cells were incubated with or without different concentrations of TFP-F1 (1, 3, and 5 μg/mL) or LPS (1 μg/mL). The levels of (**b**) TNF-*α* and (**c**) IL-6 in the RAW-Dual^TM^ KO-TLR4 cells. (**d**,**e**) Wild-type HEK293 cells and TLR4-expressing HEK 293 cells were incubated for 24 h with or without different concentrations of TFP-F1 (1, 3, and 5 μg/mL) or LPS (1 μg/mL), and the level of (**d**) TNF-*α* and (**e**) IL-6 were measured by ELISA.

**Figure 4 ijms-23-10392-f004:**
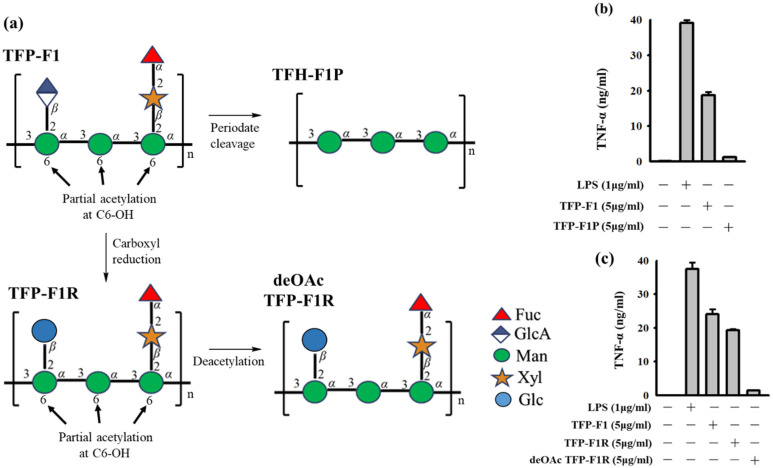
(**a**) Preparation of different TFP-F1 derivatives. TFP-F1R was prepared by treating the TFP-F1 with the CMC and NaBH_4_, while deOAc TFP-F1 was afforded by treating TFP-F1R with NaOH. TFP-F1P was the product of the periodate cleavage of TFP-F1. (**b**) The level of TNF-*α* was stimulated in the J774A.1 macrophage cells induced by the treatment of TFP-F1 of TFP-F1P. (**c**) The level of TNF-*α* secreted in the J774A.1 macrophage cells by the treatment of TFP-F1, TFP-F1R, and deOAc TFP-F1R.

**Figure 5 ijms-23-10392-f005:**
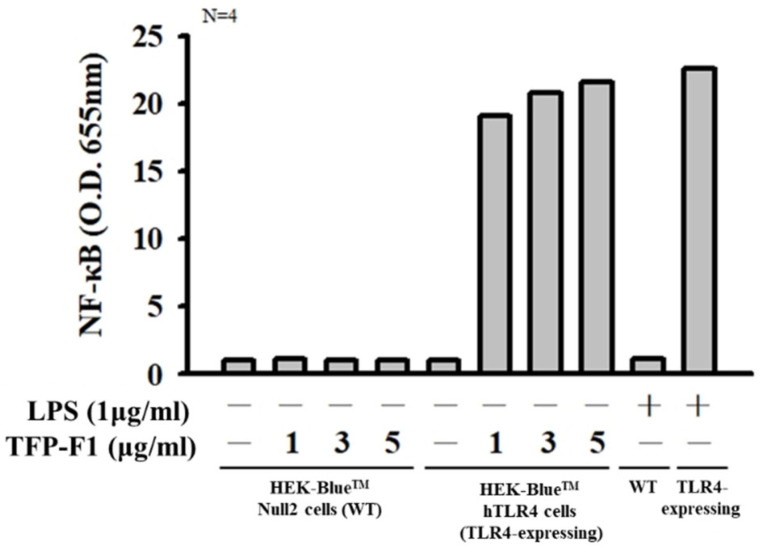
The activation levels of NF-κB were measured by an NF-κB reporter assay.

**Figure 6 ijms-23-10392-f006:**
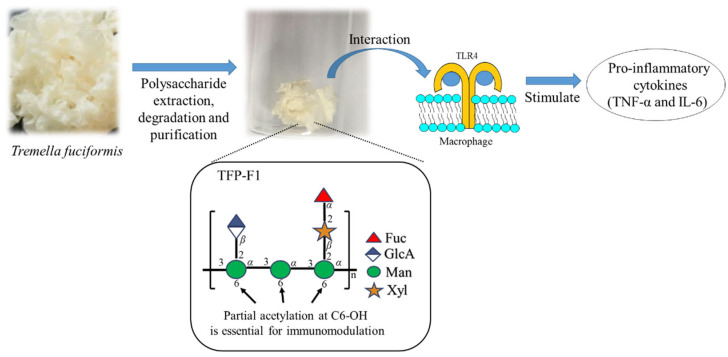
Purification of a water-soluble bioactive polysaccharide TFP-F1 from *T. fuciformis* induced the production of pro-inflammatory cytokines through interaction with TLR4. The *O*-acetyl groups at C-6 in mannose residues were essential for immunomodulation.

**Table 1 ijms-23-10392-t001:** ^1^H and ^13^C chemical shifts of TFP-F1.

Residues and Linkage	H-1/C-1	H-2/C-2	H-3/C-3	H-4/C-4	H-5/C-5	H-6/C-6
**A**. →2)*-β-*D*-*Xyl*p**-*(1→	4.37/101.7	3.50/74.1	3.59/77.7	3.55/71.8	3.94; 3.21/65.1	-/-
**B.***-β-*D-GlcA*p**-*(1→	4.44/102.2	3.33/72.6	3.41/75.3	3.53/71.8	3.65/77.0	-/175.7
**C.** →2, 3)*-α-*D*-*Man*p**-*(1→	5.06/101.3	4.15/78.1	3.94/76.1	3.70/66.3	3.75/73.5	3.86; 3.46/61.6
**D.** →3)*-α-*D*-*Man*p**-*(1→	5.10/102.3	4.17/69.9	3.94/79.1	3.71/66.3	3.78/73.6	3.82; 3.70/61.4
**E**. →2, 3)*-α-*D*-*Man*p**-*(1→	5.13/100.7	4.23/77.9	4.03/77.2	3.77/66.4	3.78/73.6	3.78; 3.78/60.6
**F.** -*α*-L-Fuc*p*-(1→	5.47/97.4	3.72/68.2	3.78/69.4	3.71/72.2	4.29/66.6	1.15/15.7

## Data Availability

Not applicable.

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
