# Peer review of "An Immunological Polysaccharide from Tremella fuciformis: Essential Role of Acetylation in Immunomodulation"

_ijms, 2022, doi:10.3390/ijms231810392_

Round 1
Reviewer 1 Report
This paper describes the immunomodulatory activity of polysaccharide from mushroom and the complete structure of that polysaccharide is proposed. The contents of this paper is interesting because it may provide the evidence to the traditional use of mushroom extract as medicines. However, the structural investigation using long-range coupling spectra of NMR in supplementary data is too complicated and hard to follow. The explanations should be simplified for the general readers of the journal. About the immunological activity of the polysaccharide, the authors should discuss more on the recognition mechanisms of the polysaccharide by TLR4, because the molecule is far different from the typical ligand of TLR4 such as lipid A. They should also discuss on the possible contamination of lipids such as fatty acids or sterols.
minor points:
page 3, line 94 and other places: Calculation method of peak area of each sugar should be described because usually mass chromatogram of GC-MS does not show the peak area. Is it calculated by GLC analysis?
page 5, line 168 and other places: The anomeric configuration of each sugar was determined, but the resolution of the NMR spectra in supplementary data is very poor, and coupling constants are not described. How is it determined?
page 6, line 192: "Figure 3b" should be "Figure 3a". Figure 3b is the result by Raw-Dual cells.
page 6, Figure 3b: "TFP-F1(ug/ml)" is lacking.
Figure 4c and Figure 5: If there are data of de-OAc-TFP-F1 (without carboxy-reduction), they should be additionally shown here to confirm the importance of acetylation.
line 66, line 96, line 104, and other places: Abbreviations such as Vc, CMC, or TFP-F1P should be spelled out when they first appear.
page 2, line 62: "biological" may be "bioactive".
page 5, line 173: "where" may be replaced by "with".
page 5, line 188-189, and page 8, line 250-251: The meaning of the sentences is not clear. They should be improved.
Author Response
Reviewer 1:
Comments and Suggestions for Authors
- This paper describes the immunomodulatory activity of polysaccharide from mushroom and the complete structure of that polysaccharide is proposed. The contents of this paper is interesting because it may provide the evidence to the traditional use of mushroom extract as medicines. However, the structural investigation using long-range coupling spectra of NMR in supplementary data is too complicated and hard to follow. The explanations should be simplified for the general readers of the journal. About the immunological activity of the polysaccharide, the authors should discuss more on the recognition mechanisms of the polysaccharide by TLR4, because the molecule is far different from the typical ligand of TLR4 such as lipid A. They should also discuss on the possible contamination of lipids such as fatty acids or sterols.
Authors response:
Thanks for your valuable suggestion.
- About the explanations of NMR spectrum in supplementary data
Due to the physical properties (high viscosity and low water solubility), previous reports on the polysaccharides from T. fuciformis were rare in elucidating their structure by detailed analyses of NMR spectroscopic data. By using various degradation methods, we finally obtained a good sample, TFP-F1, that allowed us to collect the whole NMR data. The detailed NMR assignment supports the accurancy of its chemical structure. In section 2.3, the description of NMR spectra has been simplified. Based on the above reasons, we wish to preserve the complete NMR spectroscopic data analysis in this manuscript.
- About the recognition mechanisms of the polysaccharide by TLR4
We added the following discussion related to the recognition mechanisms of the polysaccharide by TLR4 in section 2.4.
TLR4 is a well-known receptor of polysaccharides. Lipopolysaccharide, located on the outer membrane of Gram-negative bacteria, is one of the ligands of TLR4. Its toxic component, lipid A, was reported to bind with the MD2 hydrophobic pocket and connect the dimer of TLR4s [38]. The activation of TLR4 could recruit the cytoplasmic adaptor protein TIRAP (Toll/IL-1R domain-containing adaptor protein) and MyD88 (myeloid differentiation primary response gene 88) proteins to form a TLR complex, which led to the induction of pro-inflammatory cytokines, such as TNF-α and IL-6 [39,40]. However, fungus polysaccharides without lipid A were reported to trigger the immune response [25,26]. Additionally, various polysaccharides with different structures were found to be a ligand of TLR4, suggesting that the configuration and components of polysaccharides could be considered important roles in affecting TLR4 reaction [23]. (Lines 199–208)
- About the possible contamination of lipids such as fatty acids or sterols
We added some description of the LAL assay (including the procedure and results) in the revised manuscript. The results indicate that our sample, TFP-F1, is endotoxin-free, consistent with the polymycin B test. Furthermore, we confirmed that our samples were devoid of small molecules or other contamination after the various purification steps, such as precipitation, dialysis, and column chromatography. Please see the following explanation from the revised manuscript.
2.4 TFP-F1 enhances the production of pro-inflammatory cytokines through interaction with TLR4
~ Furthermore, TFP-F1 and its derivatives were quantified whether LPS existed was detected by using Limulus Amebocyte Lysate (LAL) Assay. The results (Figure S16 and Table S5) indicated that each sample (50 μL) at a concentration of 1 μg/mL contained less than 0.2 endotoxin units (EU), indicating that TFP-F1 was relative endotoxin-free (< 0.1 %).~ (Lines 187–190)
3.11 Limulus Amebocyte Lysate (LAL) Assay
The LAL assay was performed to quantify the endotoxin contamination in the TFP-F1 and its derivatives. E. coli endotoxin as standard control (25 EU), lyophilized LAL, chromogenic substrate, and LAL reagent water were purchased from Lonza (Walkersville, MD). Each sample (50 μL) with a concentration of 1 μg/mL was detected following the protocol provided by the manufacturer's instruction. (Lines 386–391)
- Conclusions
~we used a reported extraction method [45] and further purification (including precipitation, dialysis using 8–10 kDa membrane, and size-exclusion chromatography) to afford TFP-F1, a bioactive polysaccharide from T. fuciformis. In which proteins, nucleic acids, smaller molecules, and peptides were removed. (Lines 411–413)
Minor points:
- Page 3, line 94 and other places: Calculation method of peak area of each sugar should be described because usually mass chromatogram of GC-MS does not show the peak area. Is it calculated by GLC analysis?
Authors response:
Yes, the peak area of each sugar was calculated by using the GC-MS data analysis software. We added the following description in the revised manuscript.
3.6 Monosaccharide composition and linkage analysis
~The peak area of each sample was calculated through the GC-MS data analysis software.~ (Line 331)
- Page 5, line 168 and other places: The anomeric configuration of each sugar was determined, but the resolution of the NMR spectra in supplementary data is very poor, and coupling constants are not described. How is it determined?
Authors response:
Thank you very much for your suggestion. Although TFP-F1 was degraded by using the FeSO4/H2O2/ascorbic acid method, its viscosity was still very high, and its solubility in D2O was limited. It isn’t easy to collect a high-resolution 1H NMR spectrum. Consequently, we didn’t show the coupling constant of each anomeric signal. Instead, we first used the HSQC spectrum to confirm the chemical shift of each anomeric signal, and then followed the previous report (Phytochemistry, 1992, 31, 3307–3330) to confirm its configuration.
- Page 6, line 192: "Figure 3b" should be "Figure 3a". Figure 3b is the result by Raw-Dual cells.
Authors response:
Thanks for your correction. We have modified this sentence in the revised manuscript:
“Figures 3b and 3c showed a significant difference between the wild-type RAW-DualTM cells and the TLR4-knockout RAW-DualTM cells.” (Line 212)
- Page 6, Figure 3b: "TFP-F1(ug/ml)" is lacking.
Authors response:
Thanks for your suggestion. We have revised Figure 3b.
- Figure 4c and Figure 5: If there are data of de-OAc-TFP-F1 (without carboxy-reduction), they should be additionally shown here to confirm the importance of acetylation.
Authors response:
Thanks for your suggestion. Initially, we did perform the immunomodulatory activity of deOAc TFP-F1. However, this sample (deOAc TFP-F1) did not lead to the production of pro-inflammatory cytokines. Please see the figure below.
Therefore, we chose to reduce the carboxyl group first and then carried out the de-acetylation. In this way, we can evaluate the influence of both functional groups on the immune response.
- Line 66, line 96, line 104, and other places: Abbreviations such as Vc, CMC, or TFP-F1P should be spelled out when they first appear.
Authors response:
Thanks for your suggestion.
In line 67, we added the description of Vc, which first appeared in the manuscript. “The structural work was facilitated by the degradation of the full-length polysaccharide with FeSO4/ascorbic acid (Vc)/H2O2 [30].”
Also, line 98 was revised as “Reduction of the uronic acid residues in TFP-F1 was carried out by treatment with 1-cyclohexyl-3-(2-morpholino-ethyl) carbodiimidemetho-p-toluenesulfonate (CMC)~”
Furthermore, the description in line 106 was corrected as “In addition, the derivative, TFP-F1P, was prepared using the periodate cleavage of TFP-F1. This sample only consists of D-mannose (Figure S1g).”
- page 2, line 62: "biological" may be "bioactive".
Authors response:
Thanks for your suggestion. We have corrected this word. (Line 63)
- Page 5, line 173: "where" may be replaced by "with". (Line 173)
Authors response:
Thanks for your suggestion. We have corrected this word.
- Page 5, line 188-189, and page 8, line 250-251: The meaning of the sentences is not clear. They should be improved.
Authors response:
Thanks for your suggestion. The description was revised as:
“In addition, to confirm whether the sample possessed cytotoxicity against the normal cell, J774A.1 macrophages were treated with different concentrations of TFP-F1 (including 40, 20, 10, and 5 ug/mL) for three days. The viability of the macrophage cells was measured using Alamar blue assay. These results (Figure S16) showed the viability of macrophage cells is more than 85%, indicating the TFP-F1 sample is non-toxic.” (Lines 194–198)

Reviewer 2 Report
In this study, authors demonstrated that a high molecular weight bioactive polysaccharide, TFP-F1 purified from Tremella fuciformis, stimulate the secretion of TNF-alpha and IL-6. And also, the structural characters of TFP-F1 related to SAR were elucidated. There are some points to be clarified for the publication of IJMS.
1. In this study, Polymyxin B was used to exclude the possibility of LPS contamination, which is an indirect measure of the presence or absence of LPS. I wonder if there are any experimental results for LAL analysis that can quantitatively analyze the amount of LPS in samples during the preparation.
2. TNF-alpha and IL-6, as mentioned, are pro-inflammatory cytokines. TFP-F1 stimulated the secretion of two cytokines, as an evidence to stimulate immune cells via TLR4. But in other words, TFP-F1 can induce sepsis if used improperly in clinics. The related content should be included in the discussion. In addition to inflammatory cytokines, some experimental results such as the secretion of other immunomodulatory cytokines should be required to support the immunological values of TFP-F1.
Author Response
Reviewer 2:
- In this study, Polymyxin B was used to exclude the possibility of LPS contamination, which is an indirect measure of the presence or absence of LPS. I wonder if there are any experimental results for LAL analysis that can quantitatively analyze the amount of LPS in samples during the preparation.
Authors response:
Thank you very much for the valuable suggestion. In this revision, we added some description of the LAL analysis (including the procedures and results) to our revised manuscript.
2.4 TFP-F1 enhances the production of pro-inflammatory cytokines through interaction with TLR4
~ Furthermore, TFP-F1 and its derivatives were quantified whether LPS existed was detected by using Limulus Amebocyte Lysate (LAL) Assay. The results (Figure S16 and Table S5) indicated that each sample (50 μL) at a concentration of 1 μg/mL contained less than 0.2 endotoxin units (EU), indicating that TFP-F1 was relative endotoxin-free (< 0.1%).~ (Lines 187–190)
Figure S16. The standard curve of LAL assay. The E. coli endotoxin was prepared for standards of different concentrations (0.1, 0.25, 0.5, and 1.0 EU/mL).
Table S5. The endotoxin quantity of TFP-F1 and its derivatives
3.11 Limulus Amebocyte Lysate (LAL) Assay
The LAL assay was performed to quantify the endotoxin contamination in the TFP-F1 and its derivatives. E. coli endotoxin as standard control (25 EU), lyophilized LAL, chromogenic substrate, and LAL reagent water were purchased from Lonza (Walkersville, MD). Each sample (50 μL) with a concentration of 1 μg/mL was detected following the protocol provided by the manufacturer's instruction. (Lines 386–391)
- TNF-alpha and IL-6, as mentioned, are pro-inflammatory cytokines. TFP-F1 stimulated the secretion of two cytokines, as an evidence to stimulate immune cells via TLR4. But in other words, TFP-F1 can induce sepsis if used improperly in clinics. The related content should be included in the discussion. In addition to inflammatory cytokines, some experimental results such as the secretion of other immunomodulatory cytokines should be required to support the immunological values of TFP-F1.
Authors response:
Thank you very much for the invaluable comments. We added the following description to section 4. “Until now, polysaccharides isolated from Tremella fuciformis have been demonstrated to have anti-oxidative, anti-inflammatory, and anti-aging effects [50]. Also, it has been clinically used for antineoplastic agents as a safe and non-toxic natural active product [51]. Furthermore, in this study, the viability of the macrophage cells treated with TFP-F1 was detected using Alamar blue assay, and the results also indicated that this sample is non-toxic. We plan to study further the potential of endotoxin tolerance by TFP-F1 to demonstrate its immunoprotective property against bacterial infection.” (Lines 432–438)
Detail mechanism of immune responses, including cytokines stimulated by TFP-F1, will be further studied in the future.

Reviewer 3 Report
Dear authors,
I was asked to review the manuscript of Huang et al. with the title “An Immunological Polysaccharide from Tremella fuciformis: Essential of Acetylation in Immunomodulation”
The manuscript describes the authors’ study to firstly elucidate the structure of a fungal cell wall polysaccharide, secondly describe and specify immunoactivity of that polysaccharide and thirdly connect the knowledge about the activity to some structural features. In order to achieve these aims the authors used analytical tools like NMR, GC-MS, FT-IR, as well as a number of different mammalian cell lines.
Overall, the study provides detailed information and the authors’ experimental design seems to be appropriate and reasonable. This kind of research and the quality of the manuscript seems to be well-suited for the chosen journal International Journal of Molecular Sciences.
Nevertheless, I have some concerns and comments which should be answered and improved by the authors to recommend publication. I decided to recommend major revision in order to clarify the points listed below:
Major points:
1. lines 90-91: the authors mentioned that “D-glucuronic acid residues […] could not be detected by GC-MS directly without labeling in the reduction”. In my opinion this is only partly true. For the commonly used acetylation procedure it is correct, for the silylation procedure (which is used in this study) it is not correct as there are many research groups using the workflow to determine composition of aldoses and uronic acids. I understand that the authors indirectly made use of the acetylation procedure during the linkage analysis which justifies the need of the reduction step. The authors should clarify that in the text or point to literature which falsifies my argumentation.
2. I was confused by the statement “were subjected to methanolysis, acetylation and trimethylsilyation [sic!] (lines 92+93). In the “Materials and Methods” section the authors described methanolysis + trimethylsilylation but not an additional acetylation step. Why is it not mentioned? Is there any specific reason for that additional step?
3. line 311: Is it correct that the authors used 1 µg of the TFP-F1 sample in the silylation, but 1 mg in the linkage type analysis? Is this method so much more sensitive?
4. In Figure 4 the panels (b) and (c) showed discrepancy in TNF-a levels for the same sample preparation in the same concentration in the same methodology. In case of LPS and TFP-F1 there are slightly different levels which are reasonable for a biological system. In case of TFP-F1R it is around 1-2 ng/ml in panel (b) and around 20 ng/ml in panel (c). Is there any explanation for that or is it maybe a fault in data presentation? The authors should definitely comment on that.
5. I missed a few more technical details on the FT-IR measurements. It should be stated which settings were used (range which was measured, how many scan numbers…).
Minor points:
1. The subtitle of the manuscript seems to lack a word. I think the authors meant “Essential Role of Acetylation in Immunomodulation”.
2. In my opinion it would add to the text if you would shortly mention the core structure of lentinan in line 32.
3. Is the percentage in line 36 relative to dry weight?
4. It would improve readability of the sentence if you could clarify what “Vc” in line 66 stands for. By reading the methods section one would understand that it stands for vitamin C / ascorbic acid. Additionally, Figure 1a uses a different abbreviation by spelling out ascorbic acid. It should be unified throughout the text and explained before using the abbreviation.
5. lines: 192-195: The sentence structure is a little bit strange. Maybe it should be revised.
Best regards
Author Response
Reviewer 3:
Comments and Suggestions for Authors
Dear authors,
I was asked to review the manuscript of Huang et al. with the title “An Immunological Polysaccharide from Tremella fuciformis: Essential of Acetylation in Immunomodulation”
The manuscript describes the authors’ study to firstly elucidate the structure of a fungal cell wall polysaccharide, secondly describe and specify immunoactivity of that polysaccharide and thirdly connect the knowledge about the activity to some structural features. In order to achieve these aims the authors used analytical tools like NMR, GC-MS, FT-IR, as well as a number of different mammalian cell lines.
Overall, the study provides detailed information and the authors’ experimental design seems to be appropriate and reasonable. This kind of research and the quality of the manuscript seems to be well-suited for the chosen journal International Journal of Molecular Sciences.
Nevertheless, I have some concerns and comments which should be answered and improved by the authors to recommend publication. I decided to recommend major revision in order to clarify the points listed below:
Major points:
- lines 90-91: the authors mentioned that “D-glucuronic acid residues […] could not be detected by GC-MS directly without labeling in the reduction”. In my opinion this is only partly true. For the commonly used acetylation procedure it is correct, for the silylation procedure (which is used in this study) it is not correct as there are many research groups using the workflow to determine composition of aldoses and uronic acids. I understand that the authors indirectly made use of the acetylation procedure during the linkage analysis which justifies the need of the reduction step. The authors should clarify that in the text or point to literature which falsifies my argumentation.
Authors response:
Thank you very much for your suggestion. We have revised our description in section 2.2.
“In our experimental procedure of monosaccharide composition, the resulting products containing hexauronic acid could not be detected by GC-MS directly if it was not reduced.“ (Lines 194–198)
- I was confused by the statement “were subjected to methanolysis, acetylation and trimethylsilyation [sic!] (lines 92+93). In the “Materials and Methods” section the authors described methanolysis + trimethylsilylation but not an additional acetylation step. Why is it not mentioned? Is there any specific reason for that additional step?
Authors response:
Thank you very much for pointing it out. In section 3.6, “Monosaccharide composition and linkage analysis,” lines 307-308, we revised the acetylation procedure of TFP-F1 and its derivatives as follows.
“After cooling to room temperature, the reactants were evaporated under a flow of N2 gas, and then the acetylation by treatment with a solution containing 500 μL methanol, 50 μL acetic acid, and 10 μL pyridine for 20 min.” (Lines 322 and 323)
- line 311: Is it correct that the authors used 1 µg of the TFP-F1 sample in the silylation, but 1 mg in the linkage type analysis? Is this method so much more sensitive?
Authors response:
Yes, in the monosaccharide composition experiment, the concentration of each monosaccharide standard we used was 5 mM. Only 10 μL volume of each sample is required to perform this procedure. Also, it is possible to perform this experiment using the same dose of polysaccharide (1 μg). However, in the experimental process of linkage analysis, since many reagents (such as DMSO, iodomethane, pyridine, and acetic anhydrous) need to be removed, it is also necessary to use CHCl3 and H2O for multiple extractions. These extraction procedures lead to sample loss, so we prepare more samples for the reaction before the experiment begins. After many experiments, we conclude that preparing about 1 mg of a sample can allow us to complete the linkage data analysis.
.
- In Figure 4 the panels (b) and (c) showed discrepancy in TNF-a levels for the same sample preparation in the same concentration in the same methodology. In case of LPS and TFP-F1 there are slightly different levels which are reasonable for a biological system. In case of TFP-F1R it is around 1-2 ng/ml in panel (b) and around 20 ng/ml in panel (c). Is there any explanation for that or is it maybe a fault in data presentation? The authors should definitely comment on that.
Authors response:
Thank you very much for the questioning. Figure 4b is the result of the immune response in J774A.1 macrophage induced by the samples TFP-F1 and its derivative TFP-F1"P". TFP-F1P was obtained by using the periodate cleavage method of TFP-F1. The results of monosaccharide composition and linkage analyses indicated that TFP-F1P is α-1,3-mannan. However, Figure 4c showed the consequence of TFP-F1”R” induced the immune response in macrophages. TFP-F1R is the product of the reduction of TFP-F1 by CMC and NaBH4. We sincerely apologize for the misunderstanding that our writing has caused.
- I missed a few more technical details on the FT-IR measurements. It should be stated which settings were used (range which was measured, how many scan numbers…).
Authors response:
Thank you very much for your reminder. We have included the technical information about the FT-IR measurements in section 3.7. “This spectrum was recorded ranged from 4000 cm-1 to 600 cm-1. The number of scans and resolution were set up with 64 scans at a resolution of 4 cm-1.” (Lines 346–348)
Minor points:
- The subtitle of the manuscript seems to lack a word. I think the authors meant “Essential Role of Acetylation in Immunomodulation”.
Authors response:
Thank you very much for your suggestion. We followed your opinion to revise our title as “An Immunological Polysaccharide from Tremella fuciformis: Essential Role of Acetylation in Immunomodulation.”
- In my opinion it would add to the text if you would shortly mention the core structure of lentinan in line 32.
Authors response:
Thank you very much for your comment. The description of lentinan in line 32 was revised to “In addition, a β-1,3-glucan with 1,6-branches, lentinan, produced from the edible fungus Lentinus edodes, was also reported to display potential antitumor activities.”
- Is the percentage in line 36 relative to dry weight?
Authors response:
Yes, according to previous reports (J. Agric. Food Chem. 1998, 46, 4583–4586 and Int. J. Food Eng. 2018, 14. 20170288.), the percentage of the chemical composition of T. fuciformis is presented based on the air-dried weight.
- It would improve readability of the sentence if you could clarify what “Vc” in line 66 stands for. By reading the methods section one would understand that it stands for vitamin C / ascorbic acid. Additionally, Figure 1a uses a different abbreviation by spelling out ascorbic acid. It should be unified throughout the text and explained before using the abbreviation.
Authors response:
Thank you very much for pointing it out. In this revision, we unified the presence of the “FeSO4/Vc/H2O2 degradation method” in our manuscript. Also, in line 67, it was the first appearance of this method. Consequently, it was revised to FeSO4/ascorbic acid (Vc)/H2O2 method.
- lines: 192-195: The sentence structure is a little bit strange. Maybe it should be revised.
Authors response:
Thanks, we revised it.
Best regards

Round 2
Reviewer 2 Report
This manuscript can be accepted as in present form for the publication in IJMS.
Reviewer 3 Report
Dear authors,
I was asked to reassess the manuscript by Huang et al. after one round of major revision.
The authors answered all my comments sufficiently and clarified my concerns. They included more details in the methods part, very well improved the figures and also added a new experiment. Additionally, they improved the readability a lot by their corrections and explanations of abbreviations.
In conclusion, I recommend acceptance and publication in its current form. Congratulations to such an interesting and well conducted research paper!